

# Comparison of hunting site strategies of the common buzzard *Buteo buteo* in open landscapes and along expressways

Paweł Cieśluk[1], Federico Morelli[2,3] and Zbigniew Kasprzykowski[1]

[1] Faculty of Sciences, University of Siedlce, Siedlce, Poland
[2] Community Ecology & Conservation, Czech University of Life Sciences Prague, Prague, Czech Republic
[3] Institute of Biological Sciences, University of Zielona Góra, Zielona Góra, Poland

## ABSTRACT

**Background:** The expansion of human activities in their many forms increases the frequency, diversity, and scale of human-wildlife interactions. One such negative form is the expansion of road infrastructure, causing road kill and traffic-related noise as well as habitat loss and fragmentation. Even so, habitats around road infrastructure are attractive foraging areas that attract certain bird species. We assessed the impact of road infrastructure on the foraging strategies of the common buzzard *Buteo buteo*.

**Methods:** Birds were observed during two winter seasons in two land-use types, along an expressway and an open agricultural landscape. Individual birds were tracked for a 10-min sequence as a separate sample was analysed. The material, covering 1,220 min along the expressway, and 1,100 min in the agricultural landscape, was collected.

**Results:** Time spent by buzzards on medium-height sites was higher along the expressway than in farmland. Buzzards changed their hunting sites following the mean wind speed. Also, they more often changed their sites along the expressway than in farmland. The land-use types, snow cover, and the mean wind speed mediated the number of attacks on prey. These results illustrate the high plasticity of the buzzards' behaviour, which can adapt their hunting strategies to both foraging locations (expressway and farmland) and weather conditions. Roadsides along expressways are attractive foraging areas for this diurnal raptor, so reducing the risk of vehicle collisions with this and other birds of prey may require targeted planning efforts.

Corresponding author
Zbigniew Kasprzykowski,
zbigniew.kasprzykowski@uws.edu.pl

## INTRODUCTION

There is no place on Earth where humans have not directly or indirectly left their footprint (*Steffen et al., 2015*; *Williams et al., 2020*). Assessments of the impact of increasing human pressure on wildlife abound in the literature (*e.g.*, *Miller & Hobbs, 2002*; *Kerr & Deguise, 2004*; *Foley et al., 2005*; *Leu, Hanser & Knick, 2008*; *Ayram et al., 2020*; *Williams et al., 2020*; *Mu et al., 2022*), with the effects of habitat conversion typically being unfavourable to

non-human organisms and associated with declines in biodiversity (*Butchart et al., 2010*). At the same time, however, human expansion in its many forms has increased the frequency, variety, and magnitude of interactions between people and wildlife (*Sanderson et al., 2002*; *Leu, Hanser & Knick, 2008*).

The impact on wildlife of expanding road networks, especially two-lane expressways and controlled-access highways, has attracted much interest from researchers reviewed in *Pagany (2020)*, particularly in the context of their increasing density, occupied area, and vehicular traffic intensity. As network structures, road systems impact vast areas of the landscape, forming ever-larger patches of isolated habitats (*Coffin, 2007*). Roads are consistently held responsible for reducing bird populations (*Fahrig & Rytwinski, 2009*; *Benítez-López, Alkemade & Verweij, 2010*), causing the death of millions of individuals every year (*e.g.*, *Nankinov & Todorov, 1983*; *Fuellhaas et al., 1989*; *Bishop & Brogan, 2013*; *Loss, Will & Marra, 2014*). Birds are also affected by other ecological effects of roads, both direct ones such as habitat loss and fragmentation, pollution, and indirect ones like noise, artificial light, and the disruption of free movement (*Coffin, 2007*; *Parris & Schneider, 2008*; *Kociolek et al., 2011*). Road mortality and traffic noise are believed to have an impact on amphibians, reptiles, and birds, at the same time exacerbating habitat loss and fragmentation, and anticipated climate-change-driven changes in species distributions are likely to compound the overall negative effects of roads (*Hels & Buchwald, 2001*; *Steen et al., 2006*; *Andrews, Gibbons & Jochimsen, 2008*; *Woltz, Gibbs & Ducey, 2008*; *Kociolek et al., 2011*; *Heigl et al., 2017*).

On the other hand, some studies did not conclusively demonstrate a deleterious influence of the road network for birds (*e.g.*, *Palomino & Carrascal, 2007*; *Benítez-López, Alkemade & Verweij, 2010*). Furthermore, habitats associated with road infrastructure may offer attractive foraging sites and can attract certain bird species (*Fahrig & Rytwinski, 2009*; *Morelli et al., 2014*). This applies to passerines feeding on invertebrates (*Wiącek et al., 2015*) and birds of prey feeding on small mammals and carrion (*Benítez-López, Alkemade & Verweij, 2010*). The latter tend to congregate along roads when weather conditions deteriorate, especially in winter when greater concentrations of common buzzards *Buteo buteo* (*Kitowski, 2000*; *Wuczyński, 2003*; *Wikar et al., 2008*), kestrels *Falco tinnunculus* (*Mülner, 2000*; *Krasoń & Michalczuk, 2019*) and black kites *Milvus migrans* (*Meunier, Verheyden & Jouventin, 2000*) can be seen. This type of land use provides foraging conditions favourable to raptors: food is more accessible under a thin cover of snow (*Sonerud, 1986*; *Dobler, Schneider & Schweis, 1991*; *Dare, 2015*), road kill provides more opportunities for obtaining food (*e.g.*, *Orłowski & Nowak, 2006*; *Lambertucci et al., 2009*; *Borkovcová, Mrtka & Winkler, 2012*; *Lees, Newton & Balmford, 2013*; *Dare, 2015*; *Garrah et al., 2015*), and small mammals living on roadside are more numerous than on farmland (*Adams & Geis, 1983*; *Meunier et al., 1999*; *Ruiz-Capillas, Mata & Malo, 2013*). In turn, the attractiveness of roadsides to raptors contributes to their collisions with vehicles, which have been the most common reason for the anthropogenic admission to rehabilitation centers (*Maphalala et al., 2021*; *Panter et al., 2022*) and the cause of their death (*Kustusch & Wuczyński, 2023*).
Birds of prey in road and farmland foraging areas can use various hunting techniques. Unlike species actively seeking prey, such as the kestrel, the common buzzard and white-tailed eagles *Haliaeetus albicilla*, ambushes their victims with a sit-and-wait strategy (*Wuczyński, 2005*; *Lihu et al., 2007*; *Dare, 2015*; *Nadjafzadeh, Hofer & Krone, 2016*). On the other hand, the hen harrier *Circus cyaneus*, short-eared owl *Asio flammeus*, and long-eared owl *Asio otus* use energetically expensive quartering techniques during the wintering period. These species hunt in open grasslands, where prey availability is low during the snow cover period, or migrate to areas devoid of snow (*Sonerud, 1986*; *Simmons & Simmons, 2000*). Importantly, too, the road infrastructure provides a wide variety of hunting sites (*Meunier, Verheyden & Jouventin, 2000*; *Wuczyński, 2003*; *Krasoń & Michalczuk, 2019*), which when changed hunting sites increase the area inspected and thus enhances hunting success (*Kitowski, 2000*; *Cieśluk, Cmoch & Kasprzykowski, 2023*).

Open agricultural landscape is an important foraging area for wintering birds of prey. This is indicated by several studies from Europe, including Romania (*Baltag et al., 2013*, *2018*), Bulgaria (*Nikolov, Spasov & Kambourova, 2006*), Slovakia (*Nemcek, 2013*), Germany (*Dobler, Schneider & Schweis, 1991*; *Mülner, 2000*; *Schindler et al., 2012*) France (*Butet et al., 2010*), UK (*Tubbs, 1974*; *Hardey et al., 2013*; *Francksen et al., 2016*; *Walls & Kenward, 2020*) and North America (*Williams et al., 2000*; *Garner & Bednarz, 2000*). Also, in Poland, studies describing the abundance, species composition, habitat preferences, and hunting behaviour of birds of prey in agricultural areas during the wintering period have been conducted (*Kitowski, 2000*; *Kasprzykowski & Rzępała, 2002*; *Wuczyński, 2005*; *Wikar et al., 2008*; *Jankowiak et al., 2015*; *Krasoń & Michalczuk, 2019*; *Cieśluk, Cmoch & Kasprzykowski, 2023*). Monitoring of wintering raptors on roadsides was studied in Spain (*Planillo, Kramer-Schadt & Malo, 2015*) and France (*Meunier, Verheyden & Jouventin, 2000*), but roads were also often used in studies as transects from which birds of prey were counted in the open landscape of the road buffer *e.g.*, in Germany (*Gamauf, 1987*; *Busche, 1988*; *Mülner, 2000*), Czech Republic (*Ševčík, 1995*), Romania (*Baltag et al., 2013*, *2018*), Italy (*Boano & Toffoli, 2002*; *Pandolfi, Tanferna & Gaibani, 2005*), Poland (*Kitowski, 2000*; *Wuczyński, 2005*), UK (*Stevens, Murn & Hennessey, 2019*), and the United States (*e.g.*, *Olson & Arsenault, 2000*; *Bak et al., 2001*; *Ross et al., 2003*). The main focus of these studies was the common buzzard, the most common bird of prey in winter, which showed a high plasticity in its choice of foraging sites (*Kasprzykowski & Rzępała, 2002*; *Jankowiak et al., 2015*).

In this research, we assessed the impact of expressway infrastructure on the foraging strategies of the common buzzard (henceforth, buzzard). We predicted that these birds employed different hunting strategies in open farmland and near expressways. We hypothesized that the type of land use in which buzzards obtained food could contribute to differences in (1) the time spent in different categories of hunting sites, (2) the number of changes of hunting sites, and (3) the number of attacks made; a further hypothesis (4) was that the weather factors could modify the buzzards' foraging behaviour. These factors significantly influence buzzard hunting behaviour and the use of an energy-saving sit-and-wait strategy (*Cieśluk, Cmoch & Kasprzykowski, 2023*). Comparative research into the hunting strategies of this bird of prey may elucidate the plasticity of its behaviour in

response to human-induced environmental disturbances. They could explain the importance of understanding bird of prey behaviour along road infrastructure during harsh periods to enhance the conservation status of the common buzzard as an umbrella species. It may also provide a basis for targeted planning efforts to reduce the risk of collisions between birds of prey and vehicles, which can have catastrophic consequences for their occupants (see *Tschui et al., 2016*).

## MATERIALS AND METHODS

### Study areas

The research was carried out in north-eastern Poland, in two types of land-use characterized by different levels of human impact (Fig. 1). The first type was located directly on the expressway S8 (European Route E67) and was a buffer of a maximum of 50 m from the edge of the road on either side. The expressway was located between the towns of Wyszków (52.621005, 21.499630) and Zambrów (52.952531, 22.178898), over a distance of about 60 km (Fig. 1). Along about 25.3 km, the road ran through wooded areas included in the Biała Forest complex, dominated by pine monocultures. Another 29.6 km were next to open agricultural areas, and about 5.1 km were next to built-up areas, mainly in the form of residential and farm buildings located within the surrounding villages and road-related infrastructure in the form of petrol stations or catering facilities. The studied section of the expressway was a two-lane dual-carriageway. Between the carriageways, each 10.5 m wide, there was a central reservation 12 m in width. The expressway was bounded along each side by a 2.5 m tall wire mesh fence supported on metal posts inserted every 3 m. A single-lane service road ran parallel to the expressway, along which the observer moved. The verges of the roads where buzzards hunted were regularly mown during the growing season. Along the S8, buzzards perched on the wire-netting fence posts, as well as on road signs and lamp-posts, were most often located near the eight road junctions. In addition, the studied birds perched on roadside trees at the edge of forests or farmlands. No collisions between buzzards and vehicles were observed during the study. However, between 2001 and 2022, 178 cases of fatal collisions between common buzzards and vehicles were registered in Poland, which accounted for 2.78% of all bird individuals who suffered death on roads (*Kustusch & Wuczyński, 2023*). There is no data on wintering buzzards' densities along roads in the study area. However, their numbers may fluctuate due to changing weather conditions (*Kitowski, 2000*; *Wuczyński, 2003*; *Wikar et al., 2008*). The second land-use type, an area of ca 280 km$^2$ in an open agricultural landscape, was situated up to 15 km from the expressway and consisted of a mosaic of meadows, pastures, and arable land with isolated trees and shrubs. Here, buzzards reached average densities from 0.29 to 1.32 ind./km$^2$ (*Kasprzykowski & Rzępała, 2002*) and mainly chose trees and shrubs to roost.

### Field procedures

Observations in the two land-use types occurred from early November to early March during two winter seasons (2020/2021 and 2021/2022). Buzzards were observed regularly in both land-use areas approximately once a week. A total of 38 field visits were made in

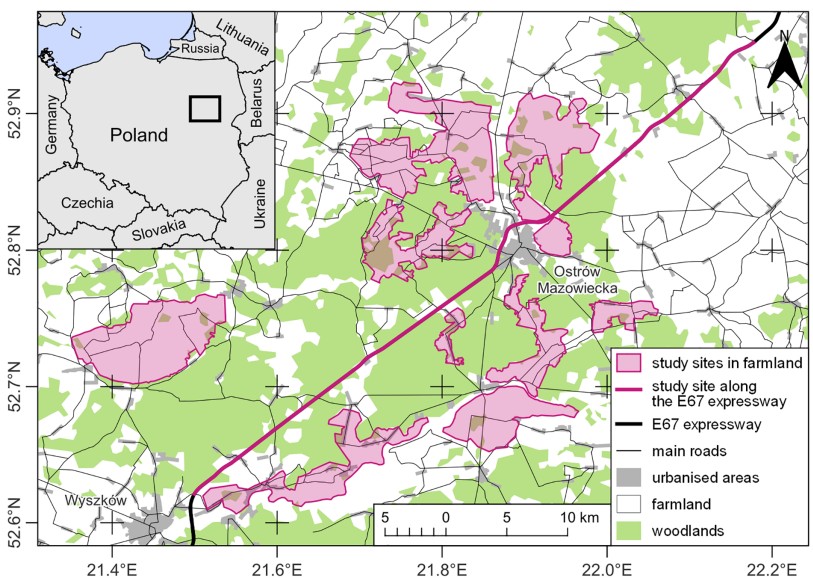

**Figure 1 Study map.** Data source: EuroStat © EuroGeographics for the administrative boundaries and the Topographic Objects Database (BDOT10k) of Head Office of Geodesy and Cartography (GUGiK). Map drawing by Zbigniew Kasprzykowski. 

the 2020/2021 season, and 33 in 2021/2022. Behavioural observations were always carried out between 08.00 and 14.00 h.

The birds' behaviour was observed from vantage points in good visibility using a 22–67 × 100 spotting scope. Each field visit involved an average of 33, 1 min of observations (range 10–70 min.). We treated each 10-min observation sequence of individual birds in the analyses as a separate sample. During this time, the number of hunting sites, and the number of attacks on prey, which were small rodents, were recorded. We did not analyze the behaviour of buzzards foraging on carrion. If a buzzard flew out of sight, we selected another bird to observe instead. We spent 2,320 min observing the buzzards, with 1,220 min spent along the expressway and 1,100 min in the agricultural landscape. Since the birds were not individually marked, some may have been observed more than once. Individuals from eastern European populations might arrive in northeastern Poland during the winter, especially when the weather conditions deteriorate in their usual wintering grounds (*Kasprzykowski & Rzępała, 2002*; *Wuczyński, 2003*; *Baltag et al., 2018*).

Following *Bohall & Collopy (1984)*, *Wuczyński (2005)*, *Bylicka et al. (2007)*, along with a few of our modifications, we considered three levels of hunting site: high–trees, lamp-posts, power poles, power lines; medium–fence posts, wire mesh posts, road signs; low–the ground (Table 1). We used a digital voice recorder to precisely measure the buzzard's time spent at hunting sites, accurately to within 1 s. We obtained mean daily temperatures (°C), mean precipitation (mm), and mean wind speeds (km/h) from the Ostrołęka weather station (52°′25′N; 22°′26′E), located 50 km from the study area. Snow cover was considered a categorical factor, either present or absent during each observation in the

**Table 1 Factors for analysing the differences in time spent on a hunting site, the number of attacks on prey, and the number of changes of hunting site by common buzzards.**

| Code | Description | Mean value (min-max) |
|---|---|---|
| Habitat | Land-use types: farmland (0) and expressway (1) | |
| TimeHigh | Time spent on a tree or a lamp-post (s) | 322.3 (0–600) |
| TimeMedium | Time spent on a fence post (s) | 180.0 (0–600) |
| TimeLow | Time spent on the ground (s) | 55.1 (0–600) |
| Attack | Number of attacks by buzzards | 0.4 (0–4) |
| Change | Number of hunting site changes | 0.9 (0–7) |
| Temp | Mean temperature (°C) | 0.37 (−18.1–10.1) |
| Wind | Mean wind speed (km/h) | 9.6 (3.5–18.5) |
| Snow | Snow cover (0-absent, 1-present) | |
| Precip | Mean precipitation (mm) | 0.80 (0–21.1) |

study area. The number of observation sequences with snow cover equaled those without snow cover (114 *vs.* 118), and the average depth of snow cover was 3.5 cm, ranging from 1 to 11.

## Statistical analyses

Three analyses were carried out using generalized linear mixed models (GLMM) in the "*lme4*" package for R (*Bates et al., 2015*). The differences in time spent on a hunting site between land-use types were modelled using binomial error variance with a logit link function. The compared pair of land-use types (habitat) was the dependent variable (binomial variable: 0 = farmland, 1 = expressway), and the global model was: Habitat ~ TimeHigh + Time Medium + TimeLow + (1|season). A second model with Poisson distribution and log link function was used to analyse four weather parameters (mean temperature, mean wind speed, mean precipitation, and snow cover) and habitat type on the number of hunting site changes. The global model for the second analysis was thus: Change ~ Habitat + Snow + Temp + Precip + Wind + (1|season). The third model, which analysed the number of attacks by buzzards, was created using a Poisson distribution and log link function. The mean temperature, wind speed, and precipitation were treated as numerical predictors. Habitat type and snow cover were the categorical factors. The global model for the number of attacks was thus the same as for the second model: Attack ~ Habitat + Snow + Temp + Precip + Wind + (1|season). Winter season was treated as a random effect in all three models. Stochastic processes were also included in the models.

In the first step, using the variance inflation factor (VIF) procedure, predictors with high and moderate multicollinearity were identified. For the first model, the variable with a high VIF value (11.7) was the category of time spent on a high site. After its removal, the remaining two predictors had a low VIF value (<1.1) and were included in the model. The VIF procedure for the other two models showed low rates (<1.1) for all variables. The predictors were, therefore, not removed from the model. In the next step, model selection

**Table 2 Results of a binomial generalized linear mixed model comparing the lengths of time common buzzards spent on the two types of hunting sites in farmland and along the expressway.**

| Variable | Estimate | SD | z | *p* |
|---|---|---|---|---|
| Intercept | −0.6316 | 0.1850 | −3.413 | 0.001 |
| **TimeMedium** | **0.0046** | **0.0007** | **6.605** | **<0.001** |
| TimeLow | −0.0001 | 0.0010 | −0.137 | 0.891 |

Note:
Statistically significant differences are indicated in bold.

was performed using the Akaike Information Criterion (AIC) (*Burnham & Anderson, 2002*). Using the "*MuMIn*" package, we calculated AICc for all possible subsets of the global model (*Bartoń, 2020*), and using model averaging based on an information criterion, we calculated a set of models that fell within the 95% confidence limits. Only the models with AIC ≤ 2 are discussed because they are treated as being equally supported (*Burnham & Anderson, 2002*). Multiple competing models were assessed concerning their fit to the data using AIC as the leading criterion, and the models with the lowest AIC value and the highest Akaike weight were selected as the best-fitting ones. The R-square for the best-supported model used the "*r.squaredFLMM()*" function from the package "*MuMIn*" (*Bartoń, 2020*). All the data were analysed in the R environment (*R Core Team, 2023*). The values reported are the mean ± 1 SE. Only those results with a I error rate of α ≤ 0.05 were assumed to be statistically significant.

## RESULTS

A comparison of the time spent at a hunting site between the land-use types showed that only the difference in hunting time on medium-height sites was statistically significant (Table 2). The model predicted that the probability of time spent on medium-height sites would be higher along the expressway than in farmland ($R^2 = 0.31$, Fig. 2).

Analysis of multiple competing models, where the number of site changes was the dependent variable, showed that four models reached a value of AICc < 2 (Table 3). These models contained two to four variables and two predictors: habitat and mean wind speed, which were the same in all four models. The mean temperature was present in three models. The three factors above formed the best model with the highest Akaike weight. Snow cover and mean precipitation were not included in the subsequent analyses. The next analysis, in which the number of attacks on prey included five models with the value of AICc < 2 (Table 3). The most recurring factors: habitat (five models), wind (four models), and snow (three models), have been included in the best model. The other two weather parameters, *i.e.*, mean temperature and mean precipitation, were omitted from the next step of the analysis.

In the best model for the number of hunting site changes ($R^2 = 0.27$), mean wind speed had a positive effect (0.08 ± 0.02), and precipitation was not significant (Table 4). Moreover, buzzards changed hunting sites significantly more often along the expressway than in farmland (0.54 ± 0.15; Fig. 3). In the third analysis, the best model ($R^2 = 0.19$) showed that hunting on the expressway increased the number of attacks on prey compared

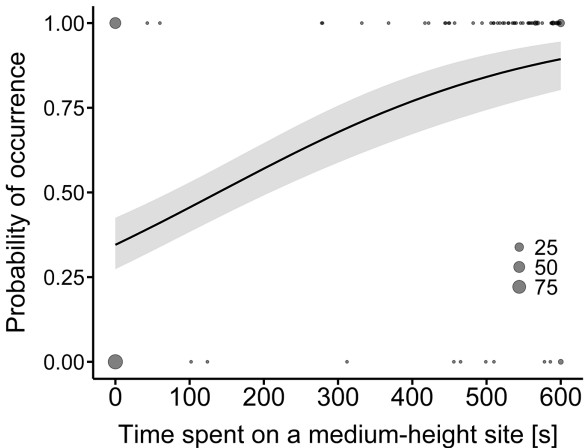

**Figure 2 Model-average estimates of the use of medium-height hunting sites by common buzzards (line) in farmland (0) and along the expressway (1).** The grey area around the line represents the 95% confidence interval. The raw data are shown by dots, the larger ones indicating a higher number of overlapping points (25, 50 and 75 cases).

**Table 3 Results of models describing the influences of habitat types and weather factors on the number of attacks and the number of hunting site changes by common buzzards.**

| Model (fixed effects) | df | LL | AIC | Δ AIC | AICwt |
|---|---|---|---|---|---|
| Number of changes | | | | | |
| Intercept+habit+wind+temp | 5 | −297.34 | 604.9 | 0.00 | 0.343 |
| Intercept+habit+wind+temp+precip | 6 | −296.91 | 606.2 | 1.25 | 0.184 |
| Intercept+habit+wind | 4 | −299.09 | 606.4 | 1.42 | 0.169 |
| Intercept+habit+wind+temp+snow | 6 | −297.26 | 606.9 | 1.94 | 0.130 |
| Number of attacks | | | | | |
| Intercept+habit+wind+snow | 5 | −179.80 | 369.9 | 0.00 | 0.228 |
| Intercept+habit+wind+temp | 5 | −180.14 | 370.6 | 0.69 | 0.161 |
| Intercept+habit+wind+snow+temp | 6 | −179.24 | 370.8 | 0.98 | 0.139 |
| Intercept+habit+snow | 4 | −181.66 | 371.5 | 1.63 | 0.101 |
| Intercept+habit+wind+precip | 6 | −179.73 | 371.8 | 1.98 | 0.085 |

Note:
Degrees of freedom (df), model log-likelihood (LL), corrected AIC criterion (AIC), difference between the model and the best model in the data set (Δ AIC), and weight for the model (AICwt) are shown.

**Table 4 Results of generalized linear mixed models of the influence of different factors on the number of changes of hunting site and the number of attacks on prey by common buzzards.**

| Fixed effects | Estimate | SE | Z value | p-value |
|---|---|---|---|---|
| Number of changes | | | | |
| Intercept | −1.2236 | 0.3004 | −4.074 | <0.001 |
| **Habitat: expressway** | **0.5371** | **0.1459** | **3.682** | **<0.001** |
| Temp | 0.0293 | 0.0161 | 1.825 | 0.068 |
| **Wind** | **0.0751** | **0.0170** | **4.502** | **<0.001** |
| Number of attacks | | | | |
| Intercept | −1.7607 | 0.4237 | −4.155 | <0.001 |

| Table 4 (continued) | | | | |
|---|---|---|---|---|
| Fixed effects | Estimate | SE | Z value | *p*-value |
| **Habitat: expressway** | **0.6534** | **0.2359** | **2.770** | **0.007** |
| **Snow: present** | **−0.6165** | **0.2552** | **−2.416** | **0.017** |
| **Wind** | **0.0555** | **0.0282** | **1.966** | **0.050** |

Note:
The reference value for snow was absence, that for habitat was farmland. Statistically significant differences are indicated in bold.

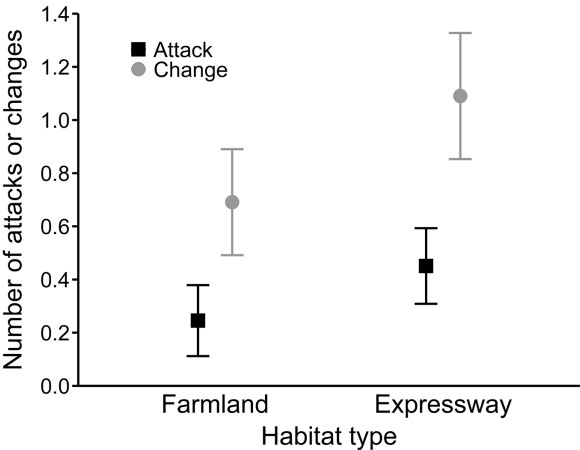

**Figure 3 Mean number of attacks (black squares) and mean numbers of hunting site changes (grey circles) by common buzzards in farmland and along the expressway.** The whiskers indicate the 95% confidence intervals.

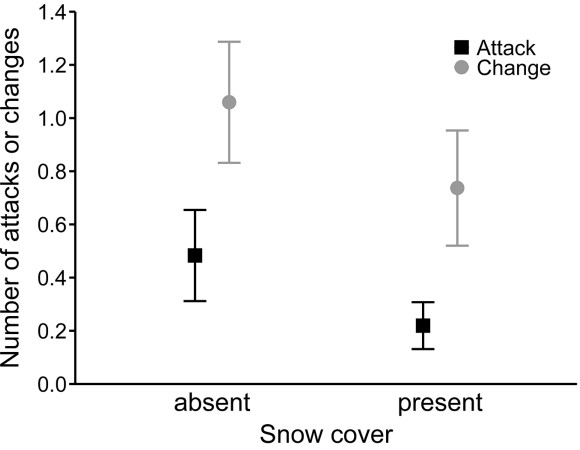

**Figure 4 Mean number of attacks (black squares) and mean numbers of hunting site changes (grey circles) by common buzzards in the presence and absence of snow cover.** The whiskers indicate the 95% confidence intervals.

to the farmland (0.63 ± 0.24; Table 4). The mean wind speed had a positive effect (0.06 ± 0.03), and the number of attacks was greater when there was no snow cover (−0.63 ± 0.25; Fig. 4).

## DISCUSSION

Our research showed that the difference in time spent hunting from medium-height sites between the two land-use types was statistically significant. Responsible for this was the availability of hunting sites, which differed in each land-use type. The farming landscape consisted of a mosaic of cultivated fields and grassland, with many balks, on which stood single trees that provided high-level sites. In contrast, both sides of the expressway were fenced off with wire netting supported on posts, which acted as medium-height sites. Our results confirm the considerable plasticity of buzzards in using various artificial structures as sites, especially if no natural ones are available in the vicinity (*Meunier, Verheyden & Jouventin, 2000*; *Mülner, 2000*; *Wuczyński, 2005*; *Dare, 2015*). The birds' use of hunting sites (lookout posts) is an energetically highly economical foraging technique that works especially well in winter. The low temperatures at this time of year and the ensuing large energy losses have to be compensated for by a suitably high biomass of victims. Otherwise, a bird may die of hypothermia (*Schmidt-Nielsen, 1997*; *Blix, 2016*). Using an optimal hunting site thus compromises the time devoted to the search for prey and the associated energy losses. The choice of such a site cannot be accidental, as it has to ensure a good field of view and rapid access to the prey. Hunting from a medium-height site may reflect this compromise, as it gives a good view of the roadside with its low vegetation (*Meunier et al., 1999*).

Using such posts can probably reduce to a minimum the energy expended on the short flight to the ground where the victim is caught (*Bryant, 1997*; *Wuczyński, 2005*). To buzzards, the possibility of surveying their hunting grounds from different heights and directions turns out to be more important than the actual type of site (*Cieśluk, Cmoch & Kasprzykowski, 2023*). However, confirmation of the choice of hunting site requires detailed research into the availability of particular types of hunting sites. It cannot be excluded that buzzards were less likely to use high-level sites on roadside trees, lamp-posts, and power lines because of the greater amount of energy needed to hunt from such a height. By contrast, the small number of buzzards hunting from medium-height sites in the farming landscape is because there are few such sites. In the Polish agricultural landscape, fence posts are found mainly on meadows, which make up less than half the area of our study plot. That is why buzzards usually hunt from single trees in areas dominated by arable fields. But again, on fields with tall crops like rapeseed, hunting from trees offers a superior view of the local landscape than doing so on the ground (*Bylicka et al., 2007*; *Wikar et al., 2008*).

The number of changes in the buzzards' hunting sites differed between the expressway and farmland. Perch hunting appears to be an advantageous hunting technique, especially when combined with short-distance movement, which allows scanning of a larger foraging area (*Kitowski, 2000*). A similar description of the hunting technique is given by *Dare (2015)*, where one of the observed buzzards systematically moved along a row of fence posts, one way and back again during the winter. The consequence of short-distance movement is that buzzard surveys their hunting grounds more efficiently and thus can attack prey more frequently (*Cieśluk, Cmoch & Kasprzykowski, 2023*). Be that as it may, it

turned out that buzzards changed their hunting sites very much more often along the expressway than in the open farming landscape. We think the key to solving this problem may lie in these two land-use types, in which the birds employ different, energetically optimal hunting strategies. It pays buzzards hunting along the expressway to make short, more frequent patrol flights. They are aided by the greater numbers of medium-height hunting sites in the form of closely separated fence posts along the expressway. By contrast, the trees in the farming landscape from which buzzards hunt are much farther apart, usually by more than 100 m. Hence, to save energy, buzzards do not make so many flights in that land-use type, especially in frosty winter weather (*Bryant, 1997*). Maintaining the right energy balance is more important than expanding the area of the hunting ground to be surveyed (*Schmidt-Nielsen, 1997*; *Blix, 2016*). It cannot be excluded that the more frequent movement of buzzards is due to a low density of rodents. There is no data on the occurrence of potential prey in both land-used types. However, in other European areas, roadsides are a more suitable refuge habitat for the most abundant small mammals (*Sabino-Marques & Mira, 2011*), which are the main component of the winter diet of buzzards (*Kowalski & Rzępała, 1997*; *Dare, 2015*; *Francksen et al., 2016*).

The weather factors affecting the buzzards' hunting strategies are snow cover and wind. The negative effect of the former on the number of their attacks on prey may be because they are harder to find in such conditions (*Sonerud, 1986*; *Dobler, Schneider & Schweis, 1991*). According to *Kowalski & Rzępała (1997)* the primary component of the winter diet of buzzards foraging in open agricultural landscapes in Poland is common voles *Microtus arvalis*, which accounts for 90% of the number of prey. However, no such data exists on the food obtained while foraging along roads. A similar relationship between snow cover and the number of attacks was found for rough-legged Buzzards *Buteo lagopus* wintering in meadow habitats (*Cieśluk, Cmoch & Kasprzykowski, 2023*). However, common buzzards are generally more sensitive to the weather, and when the snow lies thick on the ground, they change their hunting sites less often move to higher hunting sites (*Cieśluk, Cmoch & Kasprzykowski, 2023*). From there they can scan a larger area, improving their hunting efficiency and compensating for the limited prey access (*Wikar et al., 2008*). The importance of snow may decrease in the future due to fewer and fewer days with persistent snow cover. Buzzards have been found to concentrate along roads when there is a thick snow cover; on days without snow cover, there are significantly fewer buzzards along the road (*Kitowski, 2000*). Translocation of buzzards to the vicinity of roads during periods of adverse weather conditions has been demonstrated in common buzzard (*e.g., Wuczyński, 2003*; *Wikar et al., 2008*) as well as in Rough-legged Buzzards (*Watson, 1986*) and Red Kite (*Vinuela, 1997*).

The energy efficiency of raptors may also explain the positive effect of wind speed on the number of times they change their hunting site and the number of their attacks on prey because when winds are favourable, they make use of less energy-demanding flight techniques like hovering and gliding (*Rijnsdorp, Daan & Dijkstra, 1981*). However, the importance of wind is much greater for the number of change hunting sites than the number of attacks. It seems that higher wind speed helps cover distances between hunting

sites, and the greater the number of hunting sites changes, the greater the number of attacks (*Cieśluk, Cmoch & Kasprzykowski, 2023*).

## CONCLUSIONS

These results illustrate the plasticity of buzzard hunting behaviour, which can adapt to a relatively new area-expanding foraging site such as an expressway. Roadsides along expressways are attractive foraging areas, and the behaviour of buzzards foraging on expressways differs from hunting behaviour on farmland. Our findings indicate that the use of posts (medium-height hunting sites) and their frequent changes on expressways can increase the risk of collisions with vehicles. However, this study requires further analysis of the availability of hunting sites, attack success, and prey density.

## ACKNOWLEDGEMENTS

We thank Peter Senn for the English language editing and Przemysław Obłoza for help in statistical analyses.

### Funding
The study has been supported by University of Siedlce. The funders had no role in study design, data collection and analysis, decision to publish, or preparation of the manuscript.

### Grant Disclosures
The following grant information was disclosed by the authors:
University of Siedlce.

### Competing Interests
The authors declare that they have no competing interests.

### Author Contributions
- Paweł Cieśluk conceived and designed the experiments, performed the experiments, analyzed the data, prepared figures and/or tables, authored or reviewed drafts of the article, and approved the final draft.
- Federico Morelli analyzed the data, authored or reviewed drafts of the article, and approved the final draft.
- Zbigniew Kasprzykowski conceived and designed the experiments, performed the experiments, analyzed the data, prepared figures and/or tables, authored or reviewed drafts of the article, and approved the final draft.

### Data Availability
The dataset supporting reported results is available at the Mendeley Data Repository: Kasprzykowski, Zbigniew (2024), "Behavioural adaptations of the buzzard for foraging along the expressways", Mendeley Data, V2, doi: 10.17632/k5bkhxwtty.2.

## Supplemental Information

Supplemental information for this article can be found online at http://dx.doi.org/10.7717/peerj.18045#supplemental-information.

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
