# Peer review of "Comparison of hunting site strategies of the common buzzard Buteo buteo in open landscapes and along expressways"

_PeerJ, doi:10.7717/peerj.18045_

## Round 0.1 · original submission · Major Revisions

· Academic Editor

Major Revisions

Your manuscript has been now seen by three referees - and although they all agree the paper is interesting and valid, they raise major concerns (especially about the unaccounted-for factors that may have biased final outcomes). I invite you to take these points into account and revise the text accordingly.

**Language Note:** The review process has identified that the English language must be improved. PeerJ can provide language editing services - please contact us at [email protected] for pricing (be sure to provide your manuscript number and title). Alternatively, you should make your own arrangements to improve the language quality and provide details in your response letter. – PeerJ Staff

·

Basic reporting

no comment

Experimental design

no comment

Validity of the findings

no comment

Additional comments

The manuscript brings back an interesting comparative examination of the hunting behavior of the Common Buzzards observed in an agricultural landscape and along a busy expressway. The analyses based on sufficiently large dataset that consist focal bird observations obtained during two winter seasons (2020/2021 and 2021/2022) in NE Poland. In the interpretation of the results, attempts were made to link hunting behavior with the type of environment and selected weather variables. To a large degree this is a well-designed study that was adequately analyzed and concisely reported. Because it brings some new knowledge to the growing problem of the impact of road traffic on biodiversity, and fits within the scope of the PeerJ, I believe it can be published.

I note several minor criticisms connected with manuscript design and data presentation. Although brief presentation is advantageous, some parts could provide more details to facilitate the reception.

General comments
The work is based on a comparison of hunting behavior near the highway and in an open landscape. Similar samples were collected in both types of environments and all the results were arranged in the form of a comparative analysis. However, both the title of the ms and the entire Introduction focus almost exclusively on foraging along highways. I think it needs to be balanced.
Some information is missing that would give a better idea of the research area and the collected materials. The study area is the coldest region of Poland, with relatively harsh winters. It is worth characterizing both winter seasons in more detail. In particular, this concerns snow cover. Although this is a categorical variable, the reader should know what the snow depth range was. I propose to add columns in Table 1 informing about the range and average values for the presented factors. Providing raw data in a supplement is insufficient.
It is also worth providing information (your own or from the literature) on the number of buzzards wintering in the studied area, in both compared environments. In the case of highways - were there cases of buzzard collisions with vehicles observed/ how often?
Minor comments
L. 105 – please explain whether observations of the same individual were repeated within 30-minute sessions (e.g. 3x10 min). If so, this raises concerns about the independence of such 10-minute samples.

L.116 – “We recorded the time a buzzard spent on a particular behaviour”. Could you explain what types of behaviour did you note?
L. 118 – precipitation in cm3 or in mm, as stated in Table 1?
l.118 – what was the average distance of the weather station from the research area?
L. 129 – please clarify: „number of hunting site changes” instead of “changes of site”
L. 200, 217, 234 – the reference Cieśluk et al. 2023 is not listed in the reference list.
L. 214 – change “It seems, that this is the buzzards’ preferred hunting technique” to “It seems, that perch hunting is the buzzards’ preferred hunting technique”
L. 238-241 – The explanation of the effect of wind on the number of attacks and hunting site changes seems far-fetched, as the buzzard rarely hunts from the air in winter, especially near roads. Have you seen such attacks at all? I would rather look for the interaction of wind with other weather variables. Such interactions are not included in the analyzes.
Fig. 3 and 4 - The similarity of the results in Figs 3 and 4 suggests that the number of attacks were included in the number of hunting site changes. Were they? If so, it would probably be worth checking the hunting site changes that were not related to attacks.

Supplementary materials – How to understand the records with zero time spent in any hunting site? Were they flying birds during 10-min. samples? In that case, why is the number of hunting site changes listed?

Reviewer 2 ·

Basic reporting

Dear Authors,

This is an intriguing analysis on the behavioral adaptations of the Common Buzzard (Buteo buteo) for foraging along expressways, particularly pertinent in today's context of increasing road development and habitat fragmentation, especially noticeable in Eastern Europe where road infrastructure is still expanding. Additionally, with winters becoming increasingly harsh (or potentially so, given recent years), there is a significant impact on the environment, necessitating information to inform adaptive strategies for wildlife conservation.
Please check the reference list. There are some articles cited which are not in the Reference list (e.g., Cieśluk et al. 2023).

The article is written in clear English, but with some ambiguous terms, such as "habitat" or "close vicinity of the road." The authors should adhere to professional standards of clarity and expression.

The article did not include all the relevant literature on road wintering monitoring of buzzards, so the authors could improve this aspect by incorporating more studies and background information to demonstrate how their work contributes to the broader field of knowledge. Relevant prior literature should be properly referenced.

The structure of the article follows an acceptable format of standard sections. Significant deviations from this structure should only be made if they significantly improve clarity or adhere to discipline-specific customs.

Figures are relevant to the content of the article and have sufficient resolution, but the authors could improve the captions to ensure they are self-explanatory.

Experimental design

The authors present an interesting perspective, but I have concerns regarding potential biases in the analysis due to unconsidered factors or variables. Consequently, there are several questions that arise, and I will outline them below. Without addressing these aspects, it is challenging to assess the accuracy of the statistical analysis and whether the results truly reflect the species' ecology, particularly in terms of hunting behavior. I will attempt to group these questions, but some aspects may be better presented separately. I apologize for the large number of questions, but it is crucial to thoroughly examine and explain each aspect to ensure a comprehensive understanding.
The analyses of perch height are very interesting because there is a genuine debate over whether birds will use shorter perches, resulting in lower survey area coverage but less effort for attack attempts, or if they will use higher perch structures, resulting in a larger survey area but requiring more energy to fly to and from a potential prey. However, have you conducted an analysis of the available perch height in the study area? If there are more medium-sized perches near highways and fewer in agricultural areas, the birds may be opportunistic in their perch selection rather than using them strategically. Without analyzing perch size availability, it is difficult to exclude random perch use. While I have read your discussion about perch availability in the Discussion section, without testing perch availability, we cannot determine if high perch selection is a matter of choice or simply a result of perch availability. This aspect is one of the main components of this manuscript, so we cannot accurately assess the foraging strategy of the Common Buzzard if your analysis does not account for this major variable, which could affect your results. I suggest including this variable perch size availability) in your analysis to ensure that you are detecting species behavior rather than site availability.
Do you have any data on the main prey species during the winter season for the Common Buzzard? Are there any available data on food availability in these two areas? I think it would be interesting to discuss this issue, at least in the discussion section, if you do not have field data on species prey availability.
Can you provide more information about the type/structure of the study area? I'm not sure if the term "habitat" is correct; perhaps "land-use" would be more appropriate. What does "located in the direct vicinity of the expressway" mean? How far from the expressway S8 do you still consider as falling into this category? The authors should provide more details about the sections of the study area.
For the first GLM analysis, could you please add the model code? It's a bit unclear how the model was structured.
Do you have any data on successful attacks? If the birds have more missed attacks, they will have more time to make another attempt. However, if they are more successful on farmland, then it's not as necessary to move around so much and make as many attempts. Why do you think that this is the "preferred hunting technique" for the Common Buzzard? Perhaps the study area has a low density of rodents, and they need to move around quite a bit. Why do you consider that the areas closer to the expressway have a "greater availability of hunting sites"? Do you have any data on prey species density from this site, either from Poland or from other countries? If an individual goes to feed on carrion, do you consider it an attack? So, what constitutes an attack?
Also, do you have any data, from the present study or from a previous one (or from other studies in Poland, maybe in the same area) on the Common Buzzard density? Are they more present near the expressway? Is there any relation with the vicinity of the roads in your area?
As the authors presented in Table 3, there are some models that can be used, with a low Δ AIC (<2). Can you explain these models? Are they so similar? Why are they so similar? Can you comment on this subject?

The research question is well defined, relevant, and meaningful, but the authors could improve their argumentation for the study. They could explain the importance of understanding bird of prey behavior along road infrastructure during harsh periods to enhance the conservation status of the Common buzzard, as an umbrella species. While the research question is clearly defined, the variables included in the analysis could be further improved and justified.

The investigation should incorporate other variables such as perch availability, attack success, and possible prey density to achieve a high technical standard. The research was conducted in accordance with prevailing ethical standards in the field.

Validity of the findings

As mentioned in previous sections, the study is very interesting and holds potential value for species conservation. However, the authors should consider certain variables to ensure that their results are not influenced by environmental availability structures. Therefore, there are aspects that need improvement in the analysis to eliminate biases.
The conclusions are not well supported in the analysis, but this could be enhanced by incorporating new variables or conducting additional analyses to improve the study.

Additional comments

Minor comments:
Line 53: What do you mean by "poisoning" when you speak about road impact on birds? What kind of poisoning are you referring to?
Lines 62 – 70: There are also some studies that show that raptors use road infrastructure indifferently, indicating they are not particularly attracted by this human-made infrastructure. What do you think about these studies? Maybe you should mention them to present the whole picture! Moreover, in recent years, we have observed fewer snow cover days, so they can find plenty of food in fields (meadows, agricultural land, forests, etc.).
Lines 87 – 94: Including a map would help readers locate the study site and understand its position. If the authors also include information about the habitat structure, it would provide valuable context for understanding the study area's habitat matrix.
Lines 234 – 236: Can you provide evidence for this claim? Or can you add a citation to support it?
In Tables 2 and 4, could you mark the statistically significant variables? This would make it easier to identify them in the table. In Table 2, could you mark the significant variables for each GLM model presented?

Annotated reviews are not available for download in order to protect the identity of reviewers who chose to remain anonymous.

Reviewer 3 ·

Basic reporting

Although the English used is technically correct, the clarity, especially in the Discussion section, could be improved. I strongly recommend rephrasing some sentences as outlined in the detailed comments.

The primary conclusion drawn is that buzzards prefer the highest hunting posts in farmland, whereas along expressways, they tend to spend more time at medium-height posts. This conclusion appears logical, considering the distinct availability of hunting posts specific to each habitat. Interestingly, the study finds that stronger winds do not limit the hunting capabilities of common buzzards. Moreover, birds exhibited more activity along expressways than in farmland, which may be attributed to the greater abundance of medium-height hunting spots. The authors correlate the increased activity to a trade-off between the number of patrol flights and the availability of medium-high hunting sites, which reduces energy expenditure.
The reference list is comprehensive and covers most of the known literature. Nevertheless, I suggest expanding it to include a more articles, such as those by Olsson (1958) and if possible I recommend looking into the some monographs e.g. Dare P. 2015. While in the presented Manuscript Authors the authors limit themselves to the term "short flights" as a benefit from hunitng at low perches, other sources reports that hunting from perches in winter may be most energy efficient method, but buzzards are able to glide 8-10 meters from 3-4 m high bush tops or 6 meters from 1,5 meter high perch. Dare (2015, in the book The life of buzzards. Whittles Publishing, 2015. pp – 36-48) reports that buzzards are able to glide 5 m from 0.5 m perch and 20-30 m from 4m bush.

The literature on key aspects of the work such as "hunting posts," "snow cover," and "foraging behavior" is in Polish, which presents certain limitations to Polish populations. The literature, regarding birds observed in Great Britain, is rich in descriptions of the buzzard's behavior in the winter period, its mortality, diet composition, and energy requirements. I suggest to improve the reference list.
Additionally, there is a typo in the DOI for the reference Sonerud, G.A. (1986), where an extra space prevents the link from working correctly. The work by Cieśluk et al. (2023) is missing from the reference list, so I recommend a double-check of all references.

I am good with the article structure, figures and tables except for the shortcomings that I mentioned in detailed comments.

Experimental design

The research question meets the criteria of the Aims and Scope of the Journal.

Research question is clear, however the investigated knowledge gap should be identified. Buzzards hunting along expressways capture the attention not only of ornithologists but also others while traveling by car. It is satisfying that finally, someone has decided to investigate and describe this phenomenon. It is a significant contibution to the literature on the behavior of buzzards.

The authors highlight differences in the usage of hunting posts and foraging behavior between two types of habitats – farmland and areas along expressways, particularly in relation to the presence or absence of snow cover. While the main text is generally clear and easy to follow, some gaps need addressing before publication.
In the Introduction, before the hypotheses are stated, there is a noticeable lack of explanation about the types of habitats and their connection with the differences observed in hunting sites, including why the time spent and the number of changes in hunting sites are considered in the analysis.
The authors discuss the impact of road infrastructure on bird collisions and habitat fragmentation. However, I believe there is more context to provide. Given that the study shows buzzards predominantly using middle-height hunting posts and frequently changing posts, the potential for these birds to fly across expressways is an issue that should not be overlooked.
On the other side, Authors stated in Line 54 "Road mortality and traffic noise are believed to have a greater impact on birds than on any other taxonomic groups..." While it may align with the narrative on the impact of expressways, it's important to note that in the case of tertiary roads, the traffic may predominantly affect amphibians (e.g. https://doi.org/10.1186/s12898-017-0134-z).

The main issue with the statistical analysis is the inconsistency between the authors' claim of using the lme4 package for generalized linear mixed models (GLMMs) and the absence of any GLMMs in the provided source code. Additionally, the manuscript does not disclose any random effects in the methods section, despite the mention of GLMMs on line 127 and the use of the ‘lme4’ R package in the '#Model development' section of the R code. If the individual's ID serves as the random effect, this critical detail is omitted in the methods. Furthermore, the code does not include models that incorporate the random effect. Functions such as lmer, glmer, or nlmer from the lme4 package are appropriate for such analyses and should be employed.

Validity of the findings

Data files have been provided and are well-prepared for further replication of the analysis.

I am a bit surprised by some findings:
The statement on lines 178-179 lacks support from the results, and the conclusion on lines
238-241 need to be reconsidered. In my opinion, the referenced article (Clay et al. 2020) indicated that strong winds are beneficial for large soaring birds, not during the short glides of medium-sized birds of prey.

I have detailed more of my concerns in the Additional Comments section

Additional comments

Detailed comments:
L45: "reviewed in..."
L53: Are you referring to soil contamination?
L54-55: Please support this with literature or, at the very least, make references to amphibians and reptiles.
L61: “…feeding on small mammals and carrion." Also, in this case, 'birds of prey' is the preferred term.
L66-67: Please explain the advantages for raptors that utilize a sit-and-wait hunting strategy. The references should compare sit-and-wait predators with harriers and owls that use quartering techniques.
L69: Please rephrase to highlight the higher abundance of small mammals.
L70-73: This sentence is difficult to understand, particularly the second part. Could you please rephrase for clarity?
L87-94: Please provide a more detailed description of the study area, including the coordinates at the beginning and end of the transect, if possible. What was the buffer around the studied transect? Was the second habitat situated near an endpoint of the route? For instance, was the farmland habitat a 4.5 km buffer alongside the 60 km expressway, covering an area of approximately 280 km²? Additionally, mentioning the international name of the road (European Route E67) might be useful for the readers.
L102-105: For clarity, consider reordering the sentences between lines 102 and 105 to first describe the length of a separate sample (10 minutes) followed by the maximum observation time (30 minutes).
L106: The dataset consists of 476 samples, totaling 4760 minutes.
L108: 'However,' is unnecessary here.
L111-112: Generally, the absence of bird marks for the study's purpose is acceptable. However, this argument is invalid and unnecessary. It's possible to repeatedly observe the same birds from the east without such markings.
L117: Using “a dictaphone with an accuracy of one second” sounds odd. Instead, say, “We recorded our observations to an accuracy of one second using a digital voice recorder.”
L118: why not in mm?
L123: The 'glm' function is not part of the 'lme4' package. I am pretty sure 'glm' functions independently.
L127: The available R code is missing GLMMs.
L147: It seems like you have used the "r.squaredGLMM" function from the package MuMIn (Bartoń 2020).
L148: This is just my opinion, but I prefer to enclose the names of packages in apostrophes to avoid misunderstanding, e.g., 'performance', 'lme4'.
L155: It appears that there are no error bars in Figure 1. Did you statistically test the differences? It seems 'TimeHigh' has been removed from the former statistical analysis.
L157: Do you have this impression, or has it been tested? In both habitats, the only type of post for the category "TimeLow" was ground. In the next sentence, you wrote that Table 2 suggests no differences at all.
L161: It appears you present here the value of the Pseudo-R-squared for Generalized Mixed-Effect models obtained by using "r.squaredGLMM" function from the package 'MuMin'. Not sure if the reference (Nakagawa et al 2017) in the statistical analyses section is used properly then. It seems like in your analysis you applied only GLM, not GLMMs. If so, please justify in the "Statistical Analyses" section the usage of such a function "r.squaredGLMM in relation to general linear models.
L169: Same as above.
L172: Same as above.
L173: Could not find the estimates.
L179: Could not find such results across the manuscript. However, it becomes obvious when looking at Fig. 1.
L180-181: You simply repeat the conclusion from the previous sentence.
L192: "using AN OPTIMAL hunting site."
L198: Is there any support in the literature that flight energy expenditure may vary depending on length?
L200: This reference is missing.
L201: From the earlier description of the road habitat, I expected no trees to be observed along the expressway. Improving the description of the habitats in the methods section may help readers understand your conclusions.
L204: Please rephrase.
L213: "The number of changes in hunting sites" fits better.
L214-215: Please rephrase. It seems like, but some other Authors reports that hunting from perchers is most energy efficient method during winter.
L234: Regarding the previous sentence, does it mean that B. lagopus can better detect prey under snow cover?
L234-236: Have not seen such result in your article.
L238-241: This is an interesting finding. I was expecting that flying in strong wind would rather decrease the number of short flights and the total number of attacks. Please reconsider the reference Clay et al. 2020, I think the Authors showed that strong wind is beneficial for big soaring birds.
Figure 4: Lack of information in the results section about the model estimates for site changes and snow presence.

Annotated reviews are not available for download in order to protect the identity of reviewers who chose to remain anonymous.

---

## Round 0.2 · Minor Revisions

· Academic Editor

Minor Revisions

As you will see - the reviewers considered your revised manuscript as much improved. They pointed out none to minor small elements requiring corrections. I would be happy to accept your paper in principle, there are however some remaining small errors that can be corrected to make the final paper spotless. In the attached file, you will find some minor comments that I would like you to consider before the paper can be finally accepted. They are mostly cosmetic, and should not be too time-consuming. I invite you to address them.

·

Basic reporting

no comment

Experimental design

no comment

Validity of the findings

no comment

Additional comments

General comments:
This is the second time I review this paper and I am very pleased to see great improvements. In particular, supplementing the description of the research area and explaining the methods made the paper much more readable. Also, the Introduction and Discussion are improved, although the Introduction seems somewhat long, especially the recital of studies on buzzard in the penultimate paragraph.

The weakness of the work is still the lack of information about the availability of various perches in farmland and along the expressway, which was rightly pointed out by the second reviewer. However, I propose to accept the authors' explanation, since counting perches is almost impossible under the discussed field conditions.

As a non-native English speaker I probably shouldn't comment on this, but it seems that at least some of the additions to this revision need some linguistic polishing.

With minor revisions of these aesthetic errors I recommend this paper for publishing.

Reviewer 2 ·

Basic reporting

Language and Clarity:
The manuscript is written in clear, unambiguous, and professional English. The language used throughout is appropriate for the academic audience and ensures that the content is easily understandable.

Introduction and Background:
The introduction provides a comprehensive context for the study, clearly outlining the research problem and its significance. The background information is well-developed, establishing the rationale for comparing hunting site strategies in different environments. The literature review is thorough, with relevant and recent references that support the study's aims and hypotheses.

Structure and Standards:
The structure of the manuscript conforms to PeerJ standards and the norms of the discipline. Each section is well-organized, contributing to the overall clarity and coherence of the paper. The logical flow of information from the introduction to the methods, results, and discussion enhances the readability and impact of the study.

Figures:
The figures included in the manuscript are highly relevant to the study's objectives. They are of high quality, well-labelled, and effectively described, aiding in the clear communication of the research findings. The visual representation of data through these figures significantly supports the textual information and enhances the overall understanding of the results.

Raw Data:
The raw data has been supplied in accordance with PeerJ policy. This transparency allows for reproducibility and further analysis by other researchers, which is commendable and enhances the manuscript's credibility.

Experimental design

Original Primary Research:
Your study presents original primary research that is well within the scope of the journal. The comparison of hunting strategies of the Common Buzzard in different environments is both novel and significant, contributing new insights to the field of avian ecology and behavior.

Research Question:
The research question is well-defined, relevant, and meaningful. You have clearly stated how your research addresses an identified knowledge gap in understanding the hunting site preferences of the Common Buzzard. This focus on both open landscapes and expressways provides a comprehensive perspective on the species' adaptability and ecological strategies.

Rigorous Investigation:
The investigation has been performed to a high technical and ethical standard. Your methods demonstrate a rigorous approach to data collection and analysis, ensuring the reliability and validity of your findings. The ethical considerations taken during the study further underscore the integrity of your research.

Detailed Methods:
The methods section is described with sufficient detail and information, enabling replication of the study by other researchers. This transparency is crucial for advancing scientific knowledge and allowing others to build upon your work.

Validity of the findings

Replication and Rationale:
Your study's design and rationale are clearly stated, encouraging meaningful replication. By detailing the methods and benefits of your research, you have provided a strong foundation for future studies to replicate and build upon your findings, thus enriching the literature on avian hunting strategies.

Data Robustness:
The underlying data provided are robust, statistically sound, and well-controlled. Your meticulous approach to data collection and analysis ensures that the results are reliable and valid. The statistical rigor applied in your study strengthens the credibility of your conclusions.

Conclusions:
The conclusions are well stated and directly linked to the original research question. You have effectively limited your conclusions to the supporting results, which enhances the clarity and impact of your findings. This focused approach provides clear insights into the hunting strategies of the Common Buzzard in different environments.

Additional comments

In conclusion, your manuscript presents a well-executed and insightful study on the hunting site strategies of the Common Buzzard. The clear articulation of the research question, the rigorous investigation, and the detailed methodological description make this a valuable contribution to the field. I recommend it for publication in the journal.

Reviewer 3 ·

Basic reporting

The reference list has been updated, and the English is clear and understandable.The Authors reduced the number of figures.

Experimental design

No comment

Validity of the findings

No comment

Additional comments

All of the changes I recommended during initial review have been successfully implemented.

---

## Round 0.3 · accepted · Accept

· Academic Editor

Accept

Thank you for considering additional minor corrections. I'm happy to accept the manuscript as it is.